# Acaricidal Activity and Potential Application of *Atropa belladonna*

**DOI:** 10.3390/insects16111158

**Published:** 2025-11-12

**Authors:** Haicui Xie, Xuetao Yang, Caihong Cheng, Mengzhu Xian, Xiaolu Xiao, Xiuping Wang, Jun Li

**Affiliations:** 1Hebei Key Laboratory of Crop Stress Biology, College of Agronomy and Biotechnology, Hebei Normal University of Science and Technology, Qinhuangdao 066004, China; hcxie2008@126.com (H.X.);; 2Analysis and Testing Center, Hebei Normal University of Science and Technology, Qinhuangdao 066004, China; 3Oil Crops Research Institute, Chinese Academy of Agricultural Sciences, Wuhan 430062, China

**Keywords:** two-spotted spider mite, bioactive compounds, detoxification enzymes, botanical pesticide, natural enemy community

## Abstract

*Tetranychus urticae* is a global mite pest that inflicts extensive damage on crops, causing severe losses to agricultural production. For a long time, chemical pesticides have been the primary method used to control this pest. However, the overuse of chemical pesticides not only pollutes the environment and harms beneficial organisms but also leads to the gradual development of resistance in *T. urticae*. Botanical pesticides have emerged as a research hotspot in recent years due to advantages like easy degradability, low toxicity, and the ability to delay pest resistance. This study specifically evaluated the acaricidal efficacy of *Atropa belladonna* (AB) extract and tested its field performance in mixtures with four commercial chemical pesticides, namely imidacloprid (IMI), acetamiprid (ACE), thiamethoxam (TMX), and bifenthrin (BF). Results showed that AB extract exerts acaricidal effects by inhibiting detoxification enzyme activity in *T. urticae* and reducing the expression of detoxification enzyme-related genes; its main active components are hyoscyamine and genistein. The AB extract–BF combination yielded the best efficacy and did not disrupt the stability of natural enemy communities in the field. Thus, AB extract may be an eco-friendly acaricide with great potential, and its combination with bifenthrin provides a new strategy for sustainable control of *T. urticae*, which reduces chemical pesticide use while protecting natural enemies.

## 1. Introduction

The high toxicity and extensive use of chemical pesticides have caused a series of environmental problems, such as increased residue levels, harm to non-target organisms, beneficial microorganisms, and the emergence of pest resistance and resurgence [1,2]. Growing concerns regarding environmental safety have sparked interest in pest control methods that utilize eco-friendly botanical pesticides [3]. Botanical pesticides are typically sourced from plant extracts and are multi-component systems. This characteristic is advantageous for delaying the development of pest resistance [4]. They are effective in inhibiting numerous pests, diseases, and viruses [3,5]. Furthermore, they are quite readily available, inexpensive, rapidly biodegradable, and exhibit low toxicity toward beneficial organisms [6,7].

Active ingredients within botanical pesticides have been identified among numerous plant families, such as Asteraceae, Meliaceae, Solanaceae, Araceae, Zingiberaceae, Apiaceae, and Polygonaceae [3]. These compounds, which are predominantly secondary metabolites, including alkaloids, terpenoids, and flavonoids, exhibit pesticidal properties [8]. Consequently, research on the pesticidal activity of plant extracts can provide lead compounds for the development of botanical pesticides [9].

Solanaceae plants contain alkaloids and other bioactive secondary metabolites, a trait that makes them potential research subjects for active ingredients in botanical pesticides; *Atropa belladonna* (AB) is a typical example [10]. It is native to Europe, was introduced into China as a medicinal plant, and has been cultivated in medicinal plantations across the northern and southern regions of China [10]. Known commonly as belladonna or deadly nightshade, this toxic perennial herbaceous Solanaceae plant contains alkaloids (e.g., hyoscyamine, atropine, scopolamine) that form the core chemical basis of its medicinal value. Specifically, its leaves function as antispasmodic and analgesic agents, while roots are used to treat night sweats and exhibit mydriatic effects [10]. Two published patents demonstrated that mixed extracts of AB and other plants in specific proportions can be utilized as pesticides in China [11,12]. Previous studies indicated that AB extract showed toxic effects on *Sitobion avenae* (Hemiptera: Aphididae) and *Tetranychus urticae* (Acari: Tetranychidae) [13]. Nevertheless, there remains a lack of systematic research regarding its specific toxicity mechanism, as well as its biologically active ingredients toward mites.

*T*. *urticae* is a polyphagous mite pest that feeds on over 1100 host plants in 240 families, including fruit trees, vegetables, crops, and weeds [14]. Although traditional chemical pesticides can effectively control the development of their populations, long-term use of these pesticides has resulted in a gradual increase in their resistance [15,16]. Therefore, screening plant extracts with acaricidal effects, developing new botanical pesticides, and delaying the development of resistance are of practical significance.

In the present study, we evaluated the acaricidal activity of AB extract against *T. urticae*, determined the inhibitory effect of the extract on detoxification enzymes of *T. urticae*, and identified the acaricidal compounds from the extract. Furthermore, we performed ecotoxicological bioassays to evaluate the potential harmful effects of the AB extract on field populations of *T. urticae* and relevant beneficial arthropods. The results will provide a reference for increasing *T. urticae* control efficiency and reducing pesticide usage, as well as opportunities for the future development of botanical acaricides.

## 2. Materials and Methods

### 2.1. Plant Material and Preparation of Extract

The air-dried branches and leaves of AB (50 g) were powdered using a pulverizer and then extracted twice with 500 mL of 95% ethanol. The filtrate was concentrated using a rotary evaporator (Eyela, Tokyo Rikakikai Co., Ltd., Tokyo, Japan) at 52 °C to obtain the concentrate. The crude extract was dissolved in 95% ethanol to obtain a stock solution (1 g/mL).

### 2.2. T. urticae Cultures and Bioassay

*T. urticae* utilized in the bioassays were derived from a population maintained in the laboratory and reared on bean (*Phaseolus vulgaris*) plants under controlled conditions, specifically a temperature of 23 ± 1 °C, a relative humidity of 75 ± 5%, and a photophase of 16 h.

For the acaricidal activity of AB, a stock solution of AB was diluted with distilled water containing 2% acetone to five concentrations (20, 40, 60, 80, and 100 mg/mL), and the contact toxicity of the extract against *T. urticae* was measured using the leaf disk spray method [17]. The control sample was treated with distilled water containing 2% acetone. Twenty adult mites were randomly selected and transferred onto fresh bean (*P*. *vulgaris*) leaf disks (35 mm diameter), which were placed adaxial side up on moistened filter paper in Petri dishes (90 mm diameter), and then exposed to the aforementioned concentrations. Each concentration had three biological replicates, with 1.0 mL of solution uniformly sprayed per Petri dish using a JB-11 trigger hand sprayer (Taizhou Qiyong Agricultural Machinery Co., Ltd., Taizhou, China). A test for *T. urticae* was considered dead if it was unable to move and could not right itself when prodded with a tiny brush. The mortality of *T. urticae* was recorded at 24 and 48 h post-treatment to calculate the LC_50_ and LC_30_.

### 2.3. Detoxification Enzyme Assays

To determine detoxification enzyme activities, 200 *T. urticae* adults were collected and treated with the LC_50_ concentration of AB extract by spraying. The spraying amount and control treatment were the same as those described in 2.2. After 24 h of treatment, the adults were homogenized with 2 mL of frozen 0.04 mol/L PBS buffer at pH 7.0. The homogenates were centrifuged at 4 °C and 10,000× *g* for 15 min, and the supernatant was taken for the analysis of the activities of glutathione S-transferases (GSTs), Multifunctional Oxidases (MFOs), and carboxylesterases (CarEs) according to the kit instructions (GST-1-W, MFO-1-Y, and CARE-1-W, Suzhou Comin Biotechnology Co., Ltd., Suzhou, China). Each index was measured in triplicate.

Quantitative RT-PCR was employed to determine the expression of genes related to GSTs, MFOs, and CarEs. Using a Quick-RNA™ MiniPrep Kit (TR154-50; Zymo Research Corporation, Irvine, CA, USA), total RNA was isolated from adults that had undergone spray treatment and control treatment (as described in the enzyme activities determination). The quality and quantity of RNA were assessed using a NanoReady FC3100 spectrophotometer (LabTech Instruments Inc., Beijing, China). An aliquot of RNA (1 μg) was reverse transcribed into cDNA using UEIris RT mix with DNase-All-in-One (R2020, US Everbright, Suzhou, China). Target genes included the MFOs-genes (*CYP392A16* and *CYP392A16D8*), the GSTs-genes (*TuGSTd05* and *TuGSTd09*), and CarEs-genes (*TuCCE35* and *TuCCE3536*) [18]. α-tubulin was used as an internal control and amplified using the primer sequences described by Yue et al. [19]. Specific primers for the genes were designed based on *T. urticae* expressed sequence tag sequences using Primer Premier 5.0 [18]. All primer sequences are listed in Table A1. RT-qPCR was performed on an ABI Q6 Flex Real-Time PCR System (Applied Biosystems, Foster City, CA, USA). The PCRs were performed in 20 μL reaction volumes containing 1 μL of cDNA, 0.5 μL each of 10 μmol/L forward and reverse primers, 10 μL of StarLighter SYBR Green qPCR Mix (Universal), and 8.0 μL of double-distilled H_2_O, under the following thermal cycling conditions: 3 min at 95 °C followed by 40 cycles of 10 s at 95 °C and 30 s at 60 °C.

### 2.4. Analysis of Ingredients

A total of 100 μL of AB extract was transferred into an Eppendorf tube. Subsequently, 400 μL of the extraction solution (methanol: acetonitrile = 1:1, containing an isotope-labeled internal standard mixture) was added. The solution was vortexed for 30 s to ensure thorough mixing. The mixture was sonicated (Elam-sonic S60H, 100W, Elma Schmidbauer GmbH, Singen, Germany) for 10 min in an ice bath and maintained at −40 °C for 1 h. It was then centrifuged at 4 °C at 13,800× *g* for 15 min to separate the supernatant. The supernatant was collected and injected into a sample vial for analysis. The separation of the target compounds was achieved on a Phenomenex Kinetex C18 column (2.1 mm × 50 mm, 2.6 μm) in a UHPLC system (Thermo Scientific, San Jose, CA, USA). The mobile phase consisted of two components: solvent A was water containing 0.01% formic acid, while solvent B was a mixture of isopropanol and acetonitrile in a 1:1 (*v*/*v*) ratio. The sample tray was maintained at 4 °C, and the injection volume was set to 2 μL.

Mass spectrometric analyses were carried out using an Orbitrap Exploris 120 mass spectrometer (Thermo Fisher Scientific, Waltham, MA, USA), which was operated under the control of the Xcalibur software (version 4.4, Thermo Fisher Scientific). The detailed operating parameters of the mass spectrometer were as follows: sheath gas flow rate, 50 arbitrary units (Arb); auxiliary gas flow rate, 15 Arb; capillary temperature, 320 °C. The resolution of full MS was 60,000, whereas that of MS/MS was 15,000. The collision energy settings were adjusted to normalized collision energy (SNCE) values of 20, 30, and 40. The spray voltage was set to 3.8 kV for positive ionization and −3.4 kV for negative ionization. Six technical replicates were measured for both positive and negative ionization modes. The percentage composition was calculated using the area normalization method, assuming an equal detector response.

The pure hyoscyamine (HPLC ≥ 89%, B20700) and genistein (HPLC ≥ 89%, B21039) were purchased from Shanghai Yuanye Biotechnology Co., Ltd. (Shanghai, China). According to the above LC_50_ calculation method (Section 2.2), the LC_50_ of hyoscyamine and genistein against *T. urticae* were calculated.

### 2.5. Field Tests

A field efficacy trial was carried out by combining AB with insecticide (imidacloprid, IMI; acetamiprid, ACE; thiamethoxam, TMX; and bifenthrin, BF; provided by Hebei Guanlong Agrochemical Co., Ltd., Hengshui, China) treatments, that is, AB + IMI, AB + ACE, AB + TMX, and AB + BF treatments, which had a synergistic effect on *T. urticae* based on the results of a laboratory test [13]. The LC_50_ (24 h) concentrations (Section 2.2) of AB and the above chemical pesticides (Table A2) were prepared as compound agents at a 1:4 (*v*/*v*) ratio for the field efficacy trial, with distilled water containing 2% acetone as the solvent and the control.

A randomized block design was employed in the field tests. Naturally occurring *Chenopodium album* L. (family: Chenopodiaceae) plants colonized by *T. urticae* were randomly selected and labeled in each block. The AB extract and its combination with insecticide treatments were sprayed onto the leaves of the labeled plants, with five leaves selected from each plant, and each leaf was infested with 20–30 mites. The spraying amount and control treatment were the same as those described in 2.2. The abundance of *T. urticae* and their natural enemies (*Orius sauteri*, Heteroptera: Anthocoridae; *Amblyseius pseudolongispinosus*, Acari: Phytoseiidae; *Harmonia axyridis*, Coleoptera: Coccinellidae; *Chrysopa pallens*, Neuroptera: Chrysopidae; and *Erigonidium graminicolum*, Araneae: Linyphiidae) was directly monitored and recorded at intervals of 0, 1, 3, and 5 days post application on labeled plants with 15 min of inspection per plant each time, twice daily (9:00–11:00 a.m. and 3:00–5:00 p.m.). This procedure was repeated three times, with a total of three plants and 15 leaves in each treatment. Based on the above data, the population reduction rate and corrected efficacy of *T. urticae* were calculated using the following formulas:$$\mathrm{Population reduction rate}\left(\%\right)=\frac{\mathrm{initial count}-\mathrm{count after treatment}}{\mathrm{initial count}}\times 100\%$$$$\mathrm{Corrected efficacy}(\%)=(1-\frac{\mathrm{initial control count}\times \mathrm{experiment count after treatment}}{\mathrm{initial experiment count}\times \mathrm{control count after treatment}})\times 100\%$$

Characteristic indices of the natural enemy community were measured using the following formulas [20,21,22]:(a)Shannon–Wiener index (*H*′),$$H^{\prime }=-\sum_{i=1}^{s}p_{i}lnp_{i}$$


(b)Pielou evenness index (*J*),

$$J=H^{\prime }/\mathrm{lnS}$$




(c)Simpson Index (*C*),

$$C=\sum_{i=1}^{s}{p_{i}}^{2}$$



(d)Margalef’s richness (*D*),D=(S−1)/lnN
where *S* is the number of taxa, *p_i_* is the proportion of individuals in taxon *i*, and is estimated as (*n_i_*/N), where *n_i_* is the number of individuals in taxon *i* and N is the total number of individuals in the community.

### 2.6. Statistical Analysis

All data were analyzed using the statistics package SAS version 9.2 (SAS Institute, Cary, NC, USA). Mortality data were analyzed using probit analysis to determine LC_50_ and LC_30_. The 95% confidence limits (CL) were calculated using the same method. The *χ*^2^ value was used to measure the goodness of fit of the probit regression equation.

For RT-qPCR, the fold-changes in the expression of target genes were calculated using the 2^−∆∆Ct^ normalization method [23]. The detoxification enzyme activities, gene expression, population reduction rate, corrected efficacy of *T. urticae*, and characteristic indices of the natural enemy community (*H*′, *J*, *C*, and *D*) were analyzed using one-way ANOVA followed by the least significant difference (LSD) test. For all analyses, *p* < 0.05 was considered statistically significant.

## 3. Results

### 3.1. Acaricidal Activity of A. belladonna

The acaricidal activity of AB extract was enhanced by increasing the treatment time and concentration (Table 1). The LC_50_ values of the AB extract were 13.24 mg/mL at 24 h and 11.51 mg/mL at 48 h, and the LC_30_ concentrations were 6.13 mg/mL and 5.42 mg/mL at the respective time points.

### 3.2. Detoxification Enzyme Activities

AB extract treatment changed the detoxification enzyme activity of *T. urticae* (Figure 1). The activities of MFOs significantly decreased at 24 h (*F* = 72.45; *p* < 0.01) and 48 h (*F* = 11.59; *p* = 0.02), and the GST activities significantly decreased at 12 h (*F* = 45.54; *p* < 0.01) and 24 h (*F* = 68.33; *p* < 0.01) compared with the control treatment. However, there was no significant difference in the activity of CarEs between the control and treatment groups at 12, 24, and 48 h.

### 3.3. Detoxification Enzyme Gene Expression

AB extract treatment influenced the relative expression of detoxification enzyme genes in *T. urticae* (Figure 2). The relative expression of *CYP392A16* significantly decreased at 24 h (*F* = 9.51; *p* = 0.04) and 48 h (*F* = 31.88; *p* = 0.01). The relative expression of *CYP392A16D8* significantly decreased at 12 h (*F* = 20.64; *p* = 0.01), 24 h (*F* = 14.3; *p* = 0.02), and 48 h (*F* = 31.88; *p* < 0.01). The relative expression of *TuGSTd05* significantly decreased at 12 h (*F* = 350.22; *p* < 0.01), 24 h (*F* = 124.11; *p* < 0.01), and 48 h (*F* = 332.38; *p* < 0.01). The relative expression of *TuGSTd09* significantly decreased at 12 h (*F* = 33.74; *p* < 0.01), 24 h (*F* = 11.71; *p* = 0.03), and 48 h (*F* = 66.29; *p* < 0.01). The relative expression of *TuCCE35* increased at 24 h (*F* = 30.16; *p* = 0.01) and significantly decreased at 48 h (*F* = 81.12; *p* < 0.01). However, there was no significant difference in the relative expression of *TuCCE36* between the control and the treatment groups at 12, 24, and 48 h.

### 3.4. Ingredients Assay

Fourteen compound categories were identified in AB extract (Figure 3). Lipids, benzene compounds, and organic heterocyclic compounds constituted the majority of the samples. Although alkaloids and their derivatives, as well as benzenoids, appear in smaller proportions, they play critical roles in acaricidal activity [24]. Based on the relatively high content and relevant literature, two potential insecticidal substances were selected: hyoscyamine, which accounted for 8.79% of the total, and genistein, which accounted for 2.99% of the total [25,26,27,28,29]. Their structures are shown in Figure 3. The further acaricidal activity assay of hyoscyamine and genistein results indicated that with increasing treatment time and concentration, their acaricidal activity was enhanced (Table 2). The LC_50_ values of hyoscyamine were 334.05 mg/L at 24 h and 249.42 mg/L at 48 h, whereas the LC_30_ values were 135.07 mg/L and 114.30 mg/L at the respective time points. The LC_50_ values of genistein were 30.44 mg/L at 24 h and 16.14 mg/L at 48 h, while the LC_30_ values were 13.10 mg/L and 8.13 mg/L at the respective time points.

### 3.5. Field Efficacy for AB Extract Combined with Insecticide Treatment

The population reduction rate and corrected efficacy of AB extract and its combination with insecticide treatments against *T. urticae* were higher than those of a single AB application (Table 3). On the first day, the population reduction rate in the AB + BF treatment was significantly higher than that in the AB and control treatments (*F* = 5.43, *p* = 0.01); however, no significant difference in corrected efficacy was observed among the treatments. On the third day, both the population reduction rate and the corrected efficacy in the AB + BF, AB + TMX, and AB + IMI treatments were significantly higher than those in the AB and control treatments, respectively (*F* = 12.16, *p* < 0.01; *F* = 4.78, *p* = 0.02). On the fifth day, the population reduction rates in the AB + BF, AB + TMX, and AB + IMI treatments were significantly higher than those in the AB + ACE, AB, and control treatments (*F* = 16.51, *p* < 0.01). The corrected efficacy of the AB + BF treatment was significantly higher than that of all other treatments (*F* = 4.79, *p* = 0.02).

### 3.6. Characteristic Indexes of Natural Enemy Community

The effects of spraying AB extract and its combination with insecticide treatments on the characteristic indices of the natural enemy community are shown in Figure 4. On the fifth day, the *H*′ values of the AB and AB + ACE treatments were significantly higher than those of the AB + TMX and AB + IMI treatments, with no significant difference from the AB + BF treatment (*F* = 4.45, *p* = 0.03). Conversely, the *C* values of the AB and AB + ACE treatments were significantly lower than those of AB + TMX and AB + IMI treatments, with no significant difference from that of the AB + BF treatment (*F* = 3.83, *p* = 0.04). No significant differences in the *J* and *D* values were observed among the treatments during the observation period.

## 4. Discussion

Given the increasing global concern over the resistance of *T. urticae* to chemical pesticides [15], this study evaluated the potential of AB extract as a sustainable alternative. Integrated laboratory and field data highlight its insecticidal efficacy and safety profile, providing a foundation for future commercial development.

Pests can resist toxic substances by enhancing detoxification enzyme activities. Consequently, measuring their enzymatic activities may provide valuable insights into their resistance or susceptibility mechanisms to specific pesticides [30]. The results of this study demonstrated that AB extract reduced the activities of the detoxification enzymes MFOs and GSTs, suppressed the expression of their family genes *CYP392A16*, *CYP392A16D8*, *TuGSTd05*, and *TuGSTd09*, respectively. However, no effect on CarEs activity was observed. These findings indicate that AB extract targets MFOs and GSTs in *T. urticae*. Similarly, previous studies have shown that the sensitivity and resistance of *T. urticae* to different pesticides are reflected in significant changes in the activity of detoxification enzymes, including MFOs, GSTs, and CarEs [16,21,31,32].

To further identify the active compounds in the AB extract, we evaluated the acaricidal activities of hyoscyamine and genistein against *T. urticae*. Specifically, both compounds exhibited significantly higher acaricidal efficacy than the AB extract, attributed to their lower LC_50_ and LC_30_ values (Table 2). Previous studies have also reported that hyoscyamine is a major alkaloid constituent in the Solanaceae family, which exhibits significant suppressive effects on the growth and development of some insects [28,33]. Consistent with our findings, alkaloids have been shown to inhibit the MFO and GST enzymes in other insects [34,35].

Genistein is a flavonoid compound. Compounds of this nature demonstrate the dual-function capabilities of exerting lethal effects on insects and affording protective measures to beneficial insects, as exemplified by bees [36,37]. Genistein, a typical feeding-induced compound, also inhibits insect feeding behavior [26]. Notably, it disrupts insect growth and development while suppressing GST activity [25,27]. These findings are consistent with the results obtained in the present study and suggest that genistein may function via multiple pathways to control pest populations.

The combination of botanical and chemical pesticides not only enhances pest control efficacy but also lowers the minimum inhibitory concentration of chemical pesticides, mitigating environmental and ecological impacts [25,38]. This study assessed the acaricidal efficacy of AB combined with chemical pesticide treatments under field conditions. The population reduction rates and corrected efficacies for *T. urticae* in the AB + IM, AB + TMX, and AB + BF treatments were significantly higher than those in the AB treatment. Notably, the corrected efficacy in the AB + BF treatment was 74.03%, which was the highest among all the treatments. These findings suggest that the AB extract has the potential to substitute some chemical pesticides for controlling *T. urticae* in the field.

The impact of pesticide application on the characteristic indices of the natural enemy community is an indicator of the pesticide’s ecological safety and sustainability [39]. Within the first three days after AB combined with insecticide treatment, there were no significant differences in the characteristic indices of the natural enemy community among the different treatments. By the fifth day after application, compared with other treatments, the AB, AB + ACE, and AB + BF treatments had a higher *H*′ and a lower *C* in the natural enemy community. This indicates that the community in this state is healthy, better able to withstand external disturbances (e.g., climate change and invasive species), and maintains the stability of ecological functions [40,41]. Based on these field trial results, we recommend integrating AB extract with BF as a sustainable pest management strategy because this combination improves acaricidal efficacy compared to other treatments and reflects higher natural enemy community stability. While this recommendation is supported by our short-term (5-day) field data, which focuses on immediate acaricidal effect and initial natural enemy community stability, long-term validation is still needed to fully confirm the strategy’s sustained ecological suitability. Future studies will extend the period (e.g., 15–30 days) and supplement indices (e.g., predation rate) to validate long-term ecological safety.

Numerous studies have reported the acaricidal activity of plant extracts [42]. Unlike previous studies, which mainly focused on the direct toxicity of single plant extracts or their individual active components to mites (e.g., elevated mortality, reduced oviposition), this study innovatively combines AB extract with four commercial insecticides to develop a “synergistic efficacy–toxicity reduction” mite control system. This system addresses two key limitations: insufficient mite control efficacy of single plant extracts and the high resistance risk arising from long-term use of pure chemical insecticides. Notably, most existing studies lack in-depth analysis of the molecular mechanisms underlying plant extracts’ regulation of mite populations [42,43,44,45,46]. This study clarifies that AB extract exerts acaricidal effects by inhibiting detoxifying enzymes (MFOs, GSTs) in *T. urticae*. And this enzyme inhibition targets only mite-specific detoxification systems, thereby avoiding non-target impacts on natural enemies. This aligns with stable natural enemy community indices in the field, further confirming that the biochemical mechanism underpins both improved efficacy and ecological safety.

## 5. Conclusions

AB extract demonstrated significant acaricidal activity against *T. urticae*. Its mechanism of action primarily involves the inhibition of both the activities and gene expression of detoxifying enzymes (MFOs and GSTs) in *T. urticae*. Notably, the AB extract contained a relatively high content of hyoscyamine and genistein, both of which exhibited potent acaricidal properties against *T. urticae*. These compounds are promising lead candidates for the development of novel botanical pesticides. Furthermore, AB extract shows potential as a co-formulation agent when combined with chemical pesticides. This combined application not only enhanced the acaricidal efficacy against *T. urticae* but also reduced the required dosage of chemical pesticides while maintaining a positive impact on the natural enemy community.

## Figures and Tables

**Figure 1 insects-16-01158-f001:**
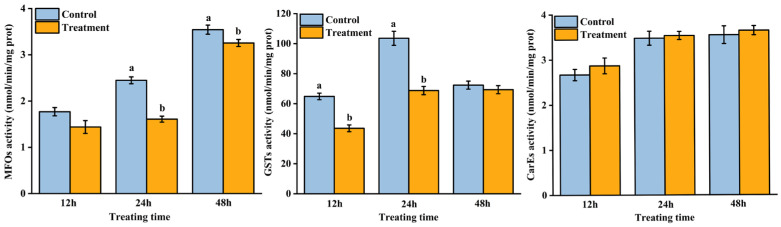
Effect of *Atropa belladonna* extract on the detoxification enzyme activity of *Tetranychus urticae* at 12 h, 24 h, and 48 h. Data are shown as the mean ± SE. Different letters represent significant differences (ANOVA, LSD test, *p* < 0.05).

**Figure 2 insects-16-01158-f002:**
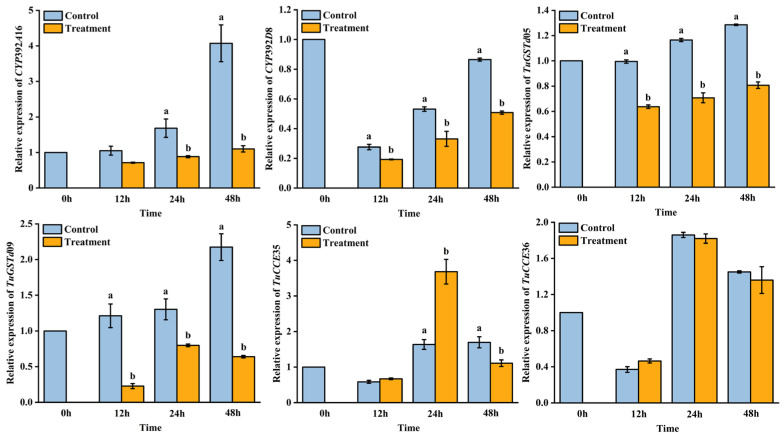
Relative expression of detoxification enzyme genes in *Tetranychus urticae*. Data are shown as mean ± SE. Different letters represent a significant difference (ANOVA, LSD test, *p* < 0.05).

**Figure 3 insects-16-01158-f003:**
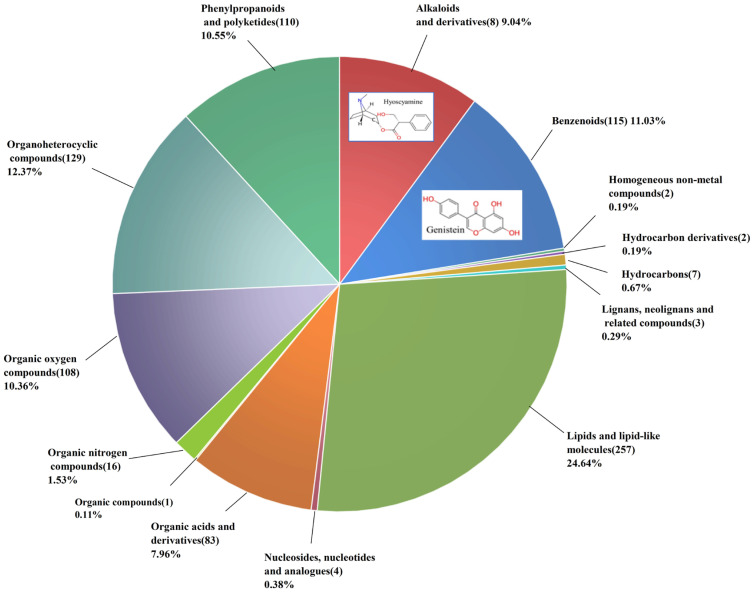
Number of chemical constituents (in brackets) and percentage of *Atropa belladonna* extract.

**Figure 4 insects-16-01158-f004:**
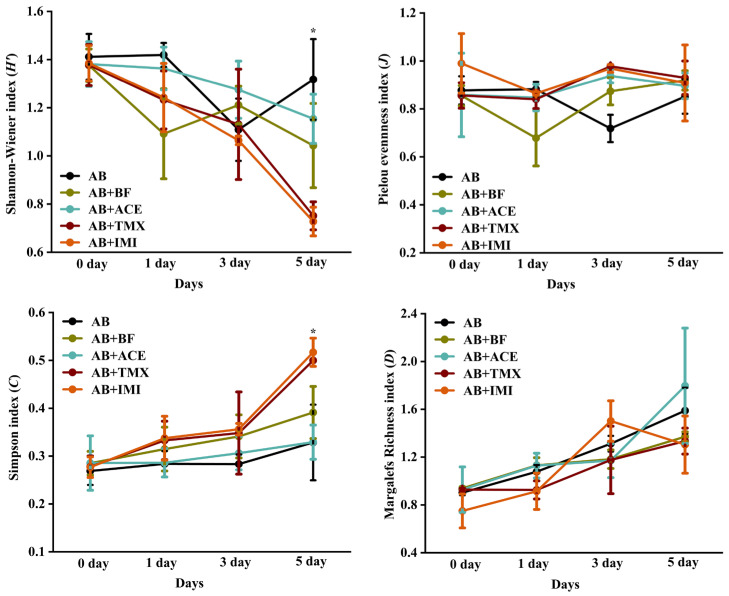
Dynamic characteristic indices of natural enemy communities among treatments after spraying *Atropa belladonna* (AB) extract, and combined with insecticide treatments. AB + IMI, *A. belladonna* and imidacloprid; AB + ACE, *A. belladonna* and acetamiprid; AB + TMX, *A. belladonna* and thiamethoxam; AB + BF, *A. belladonna* and bifenthrin. * Indicates the significant difference among treatments (ANOVA, LSD test, *p* < 0.05).

**Table 1 insects-16-01158-t001:** Acaricidal activity of *Atropa belladonna* extract against *Tetranychus urticae*.

Treatment Time	Toxicity Regression Equation (*y*=)	LC_50_ (mg/mL)	95% CL	LC_30_ (mg/mL)	95% CL	*r*	*χ* ^2^
24 h	1.568*x* − 1.759	13.24	10.52–16.13	6.13	3.68–8.15	0.97	0.22
48 h	1.604*x* − 1.702	11.51	8.93–13.97	5.42	3.16–7.32	0.97	0.43

**Table 2 insects-16-01158-t002:** Acaricidal activity of hyoscyamine and genistein against *Tetranychus urticae*.

Ingredients	Treatment Time	Toxicity Regression Equation (*y*=)	LC_50_ (mg/L)	95% CL	LC_30_ (mg/L)	95% CL	*r*	*χ* ^2^
Hyoscyamine	24 h	1.333*x* − 3.365	334.05	263.77–436.06	135.07	74.08–183.22	0.97	0.28
	48 h	1.547*x* − 3.709	249.42	195.18–304.36	114.30	66.30–154.10	0.99	2.08
Genistein	24 h	1.432*x* − 2.124	30.44	24.87–38.32	13.10	7.65–17.20	0.98	2.96
	48 h	1.761*x* − 2.127	16.14	11.91–19.54	8.13	4.50–11.20	0.99	6.29

**Table 3 insects-16-01158-t003:** Population reduction rate and corrected efficacy of *Atropa belladonna* extract and its combination with insecticide (1:4, LC_50_ concentrations) against *Tetranychus urticae*.

Combination	First Day		Third Day		Fifth Day	
	Population Reduction Rate/%	Corrected Efficacy/%	Population Reduction Rate/%	Corrected Efficacy/%	Population Reduction Rate/%	Corrected Efficacy/%
AB	23.11 ± 7.69 bc	17.91 ± 11.25 a	37.41 ± 6.46 c	25.97 ± 11.34 c	49.41 ± 4.51 b	34.45 ± 12.01 c
AB + ACE	33.43 ± 12.12 ab	30.48 ± 10.43 a	49.46 ± 8.89 bc	42.27 ± 6.74 bc	53.91 ± 7.89 b	42.67 ± 7.79 bc
AB + IMI	42.80 ± 2.61 ab	39.37 ± 3.46 a	71.77 ± 2.43 a	66.59 ± 4.91 a	72.05 ± 0.63 a	64.22 ± 4.59 b
AB + TMX	42.79 ± 6.53 ab	39.59 ± 6.03 a	62.25 ± 5.05 ab	55.76 ± 6.25 ab	73.26 ± 4.33 a	64.09 ± 7.78 b
AB + BF	46.28 ± 0.78 a	43.07 ± 2.15 a	61.98 ± 4.53 ab	55.81 ± 4.26 ab	79.21 ± 2.19 a	74.03 ± 1.11 a
Control	5.36 ± 3.84 c	—	13.65 ± 6.99 d	—	19.82 ± 8.25 c	—

AB, *Atropa belladonna* extract; AB + IMI, *A. belladonna* extract and imidacloprid; AB + ACE, *A. belladonna* extract and acetamiprid; AB + TMX, *A. belladonna* extract and thiamethoxam; AB + BF, *A. belladonna* extract and bifenthrin. Data are shown as mean ± SE. Means in the same column followed by different letters indicate significant differences among the treatment means (ANOVA, LSD test, *p* < 0.05).

## Data Availability

The original contributions presented in this study are included in the article. Further inquiries can be directed to the corresponding authors.

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
