# Peer review of "Acaricidal Activity and Potential Application of *Atropa belladonna"

_insects, 2025, doi:10.3390/insects16111158_

Round 1

Reviewer 1 Report

Comments and Suggestions for Authors

General Comments

These are my main comments on the MS entitled : “Acaricidal activity and potential application of Atropa belladonnac”. The manuscript is generally well-written, investigating the acaricidal activity of Atropa belladonna extract against Tetranychus urticae. The authors carried out lab assays and field tests in a well-structured experimental setup. My main suggestion for improvement is about the Discussion which requires substantial enrichment with other references dealing with other botanical acaricides against spider mites. I therefore recommend minor revision before the manuscript can be considered for publication.

Major Comments

Simple Summary and Abstract are well written

The Introduction could benefit from more detail on the composition and biological activity of Atropa belladonna.

Materials and Methods is very detailed and the methods are valid.

The Results are clear and well presented.

The Discussion is adequate and interprets the findings of the study. It only needs more comparison with previous relevant references on the acaricidal action of other botanicals. The addition of some references (1-2 paragraphs) would substantially improve the MS.

Minor Comments

Line 15 “to crops” → should be “on crops.”

Line 27 “in field” → “in the field.”

Line 39 “treatment ……..significantly enhanced the corrected efficacy” → “treatments …………significantly enhanced corrected efficacy.”

Line 95 Better use “crude extract” instead of “concrete”.

Author Response

Comments 1: These are my main comments on the MS entitled: “Acaricidal activity and potential application of Atropa belladonnac”. The manuscript is generally well-written, investigating the acaricidal activity of Atropa belladonna extract against Tetranychus urticae. The authors carried out lab assays and field tests in a well-structured experimental setup. My main suggestion for improvement is about the Discussion which requires substantial enrichment with other references dealing with other botanical acaricides against spider mites. I therefore recommend minor revision before the manuscript can be considered for publication.

 Response 1: We sincerely appreciate your positive comments on our manuscript’s experimental design and writing quality, as well as your constructive suggestion to enrich the Discussion. We fully accept this advice. Furthermore, we have cited recent literature to further elaborate on the Discussion, as detailed in lines 360-379.

Comments 2: Simple Summary and Abstract are well written

Response 2: We sincerely thank you for your positive feedback.

Comments 3: The Introduction could benefit from more detail on the composition and biological activity of Atropa belladonna.

Response 3: We sincerely appreciate your valuable suggestion regarding enriching the Introduction. We have accordingly revised the Introduction section in lines 72-76.

Comments 4: Materials and Methods is very detailed and the methods are valid.

The Results are clear and well presented.

Response 4We sincerely thank you for your positive feedback.

Comments 5: The Discussion is adequate and interprets the findings of the study. It only needs more comparison with previous relevant references on the acaricidal action of other botanicals. The addition of some references (1-2 paragraphs) would substantially improve the MS.

Response 5: As mentioned above, we have added content to the Discussion section in lines 360-379.

Minor Comments

Comments 6: Line 15 “to crops” → should be “on crops.”

Response 6: Thank you for your careful feedback. We revised that in line 15.

Comments 7: Line 27 “in field” → “in the field.”

Response 7: Thank you for pointing out this wording detail. We revised that in line 28.

Comments 8: Line 39 “treatment ……..significantly enhanced the corrected efficacy” → “treatments …………significantly enhanced corrected efficacy.”

Response 8: Thank you for your careful feedback. We revised that in line 40.

Comments 9: Line 95 Better use “crude extract” instead of “concrete”.

Response 9: Thank you for your careful feedback. We revised that in line 100.

Reviewer 2 Report

Comments and Suggestions for Authors

The manuscript presents a well-structured investigation into the acaricidal potential of Atropa belladonna (AB) extract against Tetranychus urticae, both in laboratory and field settings. The study is timely and relevant, given the growing need for sustainable alternatives to chemical pesticides. The combination of biochemical, molecular, and ecological approaches strengthens the work. However, several issues require clarification and improvement before the manuscript can be considered for publication.

1 The manuscript would benefit from thorough proofreading by a native English speaker or professional editing service. Several sentences are awkwardly constructed or unclear, which hinders comprehension.

Example: “The effective acaricidal properties of the AB extract make it a promising candidate and a viable and eco-friendly alternative…” (Abstract) – Repetitive and unclear.

2 Section 2.2: Please specify the number of mites used per replicate and the total number of replicates. The range “10 to 20” is too broad and may affect reproducibility.

3 Section 2.5: The field trial design should be described in more detail. How were the plants selected? Were the treatments randomized? How was the natural enemy community sampled?

4 Section 2.6: The statistical analysis section should explicitly state the post-hoc test used for ANOVA comparisons.

5 Table 3: The use of letters to denote significance is inconsistent (e.g., “bc”, “ab”, “a”). Please ensure all values are clearly marked and explained in the caption.

6 The discussion should more directly link the biochemical findings (enzyme inhibition) with the field results (efficacy and natural enemy impact).

7 The claim that “AB extract does not disrupt natural enemy communities” is based on limited indices and a short time frame (5 days). This should be tempered with caution, and the limitations acknowledged.

8 While hyoscyamine and genistein were identified as active compounds, their specific roles in enzyme inhibition (MFOs, GSTs) are not experimentally verified. Additional bioassays or molecular docking studies would strengthen the mechanistic claims.

9 The use of Chenopodium album as a host plant in field trials is unusual. Justify this choice and discuss whether results are generalizable to crop plants.

10 Introduction: The transition from botanical pesticides in general to AB specifically could be smoother.

11 Abbreviations: Define all abbreviations at first use (e.g., MFOs, GSTs, CarEs).

Author Response

Comments 1: The manuscript would benefit from thorough proofreading by a native English speaker or professional editing service. Several sentences are awkwardly constructed or unclear, which hinders comprehension.

Example: “The effective acaricidal properties of the AB extract make it a promising candidate and a viable and eco-friendly alternative…” (Abstract) – Repetitive and unclear.

Response 1: We sincerely appreciate your valuable suggestion regarding thorough proofreading of the manuscript. We fully agree that linguistic clarity is critical for effective communication of our research, and we have engaged a professional English editing service to conduct a comprehensive language revision.

Regarding the example in the Abstract: “The effective acaricidal properties of the AB extract make it a promising candidate and a viable and eco-friendly alternative to chemical pesticides for the sustainable management of T. urticae”, we have revised it to: “the AB extract represents an environmentally benign alternative to chemical pesticides for the sustainable control of T. urticae.” (lines 45-47). This revision removes redundant phrasing while retaining all key information, improving readability.

Comments 2: Section 2.2: Please specify the number of mites used per replicate and the total number of replicates. The range “10 to 20” is too broad and may affect reproducibility.

Response 2: We sincerely appreciate your valuable suggestion. The number of mites was confirmed to be 20 through experimental records. And we revised that into ‘twenty’ in line 110.

Comments 3: Section 2.5: The field trial design should be described in more detail. How were the plants selected? Were the treatments randomized? How was the natural enemy community sampled?

Response 2: Thank you for your suggestions on Section 2.5. We have supplemented details in lines 182-193.

Comments 4: Section 2.6: The statistical analysis section should explicitly state the post-hoc test used for ANOVA comparisons.

Response 4: Thank you for your suggestions on Section 2.6. We have supplemented the description of the post-hoc test methods used in ANOVA comparisons in line 221.

Comments 5: Table 3: The use of letters to denote significance is inconsistent (e.g., “bc”, “ab”, “a”). Please ensure all values are clearly marked and explained in the caption.

Response 5: Thank you for noting the inconsistent use of significance-denoting letters in Table 3. We have standardized the lettering rules for significance marking. The caption has been supplemented with the explanation: "Means in the same column followed by different letters indicate significant differences among the treatment means (ANOVA, LSD test, p < 0.05)."

Comments 6: The discussion should more directly link the biochemical findings (enzyme inhibition) with the field results (efficacy and natural enemy impact).

Response 6: Thank you for your valuable suggestion. We have revised the Discussion section to more directly establish the link between biochemical findings (enzyme inhibition) and field results (efficacy and natural enemy impact) in line 375-379.

Comments 7: The claim that “AB extract does not disrupt natural enemy communities” is based on limited indices and a short time frame (5 days). This should be tempered with caution, and the limitations acknowledged.

Response 7: Thank you for your insightful comment. We fully agree and have revised the discussion to address this point in line 360-365.

Comments 8: While hyoscyamine and genistein were identified as active compounds, their specific roles in enzyme inhibition (MFOs, GSTs) are not experimentally verified. Additional bioassays or molecular docking studies would strengthen the mechanistic claims.

Response 8: Thank you for your valuable comment on the specific roles of hyoscyamine and genistein in enzyme inhibition. We agree that verifying their individual effects via bioassays or molecular docking would deepen mechanistic understanding, which we regard as a key future direction.

Comments 9: The use of Chenopodium album as a host plant in field trials is unusual. Justify this choice and discuss whether results are generalizable to crop plants.

Response 9: Thank you for your question regarding the selection of Chenopodium album as the host plant. We chose it for three key reasons: firstly, it is a natural, common host of Tetranychus urticae in agricultural fields and often acts as a weed "reservoir" for mites that migrate to crops, making results relevant to on-farm management; secondly, its uniform growth and ease of standardization minimize host plant variability, ensuring consistent trial conditions; thirdly, our study focused on contact toxicity to mites, which is less affected by host plant species.

Comments 10: Introduction: The transition from botanical pesticides in general to AB specifically could be smoother.

Response 10: Thank you for your valuable suggestion. We agree that the transition needs improvement and have revised the introduction by adding a bridging sentence in line 68-70.

Comments 11: Abbreviations: Define all abbreviations at first use (e.g., MFOs, GSTs, CarEs).

Response 11: Thank you for pointing this out. We have revised the manuscript to define all abbreviations (including MFOs, GSTs, CarEs, etc.) at their first occurrence in line 124-125.

Reviewer 3 Report

Comments and Suggestions for Authors

Thank you for giving me the opportunity to read and review the research article titled “Acaricidal activity and potential application of Atropa belladonna”. This work is broad in scope, ranging from the extraction of plant tissues, chemical characterisation, acaricidal tests in both laboratory and field conditions, molecular biology, pesticide formulation, and ecology (just to mention a few aspects) - my congratulations to the authors for their effort.

Anyhow, I have one major concern.

When you perform the leaf disc spray method, you state that your mites are located under the leaf (line 108), which is then sprayed. Therefore, the mites do not come into contact with the treatment directly, but only indirectly when moving to the upper side. This is what I understand from your Materials and Methods. I tried to check reference 13, but it is a thesis, so I could not access it (it would be useful to add an additional, more accessible reference). As far as I know, in this method, mites should be put on the upper side of the leaf and receive the treatment directly. In that regard, here comes my doubt. For the “Detoxification enzyme assays”, you state that you treat the mites directly by spraying the AB extract at the LC50 concentration. If both procedures are described correctly, we are dealing with mites treated in two completely different ways -directly and indirectly- and, thus, not comparable. This might just be an issue with the term “underside” (line 108) used instead of “upper side”, which would make it a minor problem, but I would appreciate clarification.

Some other concerns, mainly regarding missing details from the Materials and Methods section, are listed here:

  • Whether adding the authority of plant and animal species names is up to you, as long as consistency is maintained.
  • You mention industrial alcohol (lines 94 and 96); I assume you mean ethanol, but please specify it.
  • Spraying amount (lines 110 and 116) is unclear in a scientific context. Please provide the exact volume used, as well as the mean area of the leaves used. I want to be able to understand the amount of vegetal extract present on each cm² of the leaf, so both parameters (along with the concentrations reported) are fundamental. The same applies to the amount used in the field trials (line 178).
  • What device did you use for the spraying treatments? I mean both under laboratory conditions (lines 109 and 116) and in the field (line 178).
  • How was the monitoring (line 182) of the presence of natural enemies performed? Did you record videos or conduct visual inspections? For how long? One hour at a time? More? At what time of the day?
  • What post-hoc test (or tests) did you use to highlight the differences observed by the ANOVA analysis (line 211)?
  • Figures 3 and 4 are too small to detect potential errors; I prefer not to comment on them at this stage.
  • All in all, what about combining hyoscyamine or genistein with BF or other synthetic acaricides? Would it be more expensive than using the whole AB extract? Please comment on this aspect.

Other comments and suggestions are noted in the attached PDF.

Round 2

Reviewer 3 Report

Comments and Suggestions for Authors

I would like to thank the authors for the trust they have shown in my revision. All my doubts have been clarified.

Only two minor notes:

Line 35: please add Atropa belladonna before the abbreviation (AB)

Lines 70-72: please rephrase; there is a missing verb and probably a subject

Congratulations again on your work.